# Integration of Multi-Omics, Histological, and Biochemical Analysis Reveals the Toxic Responses of Nile Tilapia Liver to Chronic Microcystin-LR Exposure

**DOI:** 10.3390/toxins16030149

**Published:** 2024-03-14

**Authors:** Yichao Li, Huici Yang, Bing Fu, Gen Kaneko, Hongyan Li, Jingjing Tian, Guangjun Wang, Mingken Wei, Jun Xie, Ermeng Yu

**Affiliations:** 1Faculty of Fisheries and Life Sciences, Shanghai Ocean University, Shanghai 201306, China; yichaoli1205@163.com; 2Pearl River Fisheries Research Institute, Chinese Academy of Fishery Sciences, Guangzhou 510380, China; yanghuici2023@126.com (H.Y.); lihongyan@prfri.ac.cn (H.L.); tianjj@prfri.ac.cn (J.T.); gjwang@prfri.ac.cn (G.W.); 3College of Marine Sciences, South China Agricultural University, Guangzhou 510640, China; fub@prfri.ac.cn; 4College of Natural and Applied Science, University of Houston-Victoria, Victoria, TX 77901, USA; kanekog@uhv.edu; 5Maoming Branch, Guangdong Laboratory for Lingnan Modern Agriculture, Maoming 525000, China; weimingken@gdupt.edu.cn

**Keywords:** microcystin-LR, nile tilapia, hepatotoxicity, bile acid biosynthesis, molecular mechanism

## Abstract

Microcystin-LR (MC-LR) is a cyanobacterial metabolite produced during cyanobacterial blooms and is toxic to aquatic animals, and the liver is the main targeted organ of MC-LR. To comprehensively understand the toxicity mechanism of chronic exposure to environmental levels of MC-LR on the liver of fish, juvenile Nile tilapia were exposed to 0 μg/L (control), 1 μg/L (M1), 3 μg/L (M3), 10 μg/L (M10), and 30 μg/L (M30) MC-LR for 60 days. Then, the liver hepatotoxicity induced by MC-LR exposure was systematically evaluated via histological and biochemical determinations, and the underlying mechanisms were explored through combining analysis of biochemical parameters, multi-omics (transcriptome and metabolome), and gene expression. The results exhibited that chronic MC-LR exposure caused slight liver minor structural damage and lipid accumulation in the M10 group, while resulting in serious histological damage and lipid accumulation in the M30 group, indicating obvious hepatotoxicity, which was confirmed by increased toxicity indexes (i.e., AST, ALT, and AKP). Transcriptomic and metabolomic analysis revealed that chronic MC-LR exposure induced extensive changes in gene expression and metabolites in six typical pathways, including oxidative stress, apoptosis, autophagy, amino acid metabolism, primary bile acid biosynthesis, and lipid metabolism. Taken together, chronic MC-LR exposure induced oxidative stress, apoptosis, and autophagy, inhibited primary bile acid biosynthesis, and caused fatty deposition in the liver of Nile tilapia.

## 1. Introduction

With the increase in extreme climate and water eutrophication, outbreaks of cyanobacterial blooms are becoming more frequent; not only do they contaminate water but they also produce extremely toxic cyanotoxins, which cause widespread public concern [1]. Microcystin-LR (MC-LR) is widely known to be the most prevalent and dangerous microcystin (MC) variations with high toxicity. 

Compared with other organisms (e.g., terrestrial animals and amphibious animals), aquatic animals (e.g., fish) are more susceptible to MC-LR, as they are first released from cyanophytes in water. MC-LR has been widely known to induce multiorgan toxicity (e.g., enterotoxicity, hepatotoxicity, and neurotoxicity) in aquatic organisms [2,3,4]. MC-LR can interfere with the dynamic balance of protein phosphorylation and dephosphorylation by inhibiting the activity of protein phosphatase 1/2A, causing protein hyperphosphorylation, leading to cytoskeletal damage, affecting the cell cycle, and triggering apoptosis [1]. MC-LR can also reduce the activity of antioxidant enzymes and cause oxidative stress in cells or tissues [3,4]. Among the organs, the liver is widely known to be the primary targeted organ of MC-LR [2]. Numerous studies have investigated the acute and subacute toxicities of MC-LR in the liver of aquatic animals [3]. Comparatively, fewer studies investigated the hepatotoxicity of chronic exposure to MC-LR on aquatic organisms. A previous study on zebrafish showed that chronic exposure to ambient levels of MC-LR (10 μg/L) led to slight structural damage, dysfunction of the mitochondria, and disturbed the lipid metabolism in the liver of zebrafish [1]. Another study on male zebrafish reported that chronic MC-LR exposure resulted in liver lipid accumulation by triggering endoplasmic reticulum stress [5]. Moreover, the liver is a key organ for the production of bile acids, which play important functional roles in maintaining liver homeostasis through regulating lipid metabolism, cholesterol metabolism, and oxidative stress [6]. Thus, the disorder of bile acid metabolism is regarded as an important factor in the induction of liver disease (e.g., fatty liver) [7]. However, the effects of long-term MC-LR exposure on bile acid synthesis in the liver of aquatic animals were rarely studied. Moreover, although the MC-LR level in natural freshwater typically ranges from 0.1 to 10 μg/L when blooms occur, the ambient concentrations of MC-LR reached sublethal levels (30 μg/L) after the collapse of a large cyanobacterial bloom in many cases [6,8]. Thus, few studies comprehensively investigated the toxic effects of higher environmental concentrations of MC-LR exposure on the liver health of fish. In this study, Nile tilapia (*Oreochromis niloticus*), a widely distributed commercial fish, was exposed to environmental concentrations of MC-LR (0, 1, 3, 10, and 30 μg/L) to comprehensively explore the toxicity mechanism of chronic MC-LR exposure on the liver of fish by combining analysis of histology, biochemical results, and multi-omics (transcriptome and metabolome), which provided new insights into the environmental health risks of MC-LR to aquatic animals.

## 2. Results

### 2.1. Biochemical Parameters of the Serum and Liver

As shown in Table 1, MC-LR exposure enhanced the levels of serum TG and HDL-C in the M10 and M30 groups, as well as TC and LDL-C in the M30 group, compared to the control (*p* < 0.05), which indicated the occurrence of hyperlipidemia. Moreover, the increased serum ALT, AST, and AKP in the M30 group suggested hepatic damage due to MC-LR exposure (*p* < 0.05).

Moreover, MC-LR exposure increased hepatic lipid metabolism parameters, as illustrated by increased contents of TG and LDL-C in the M10 and M30 groups, as well as TC and HDL-C in the M30 group (*p* < 0.05) (Table 2). Additionally, MC-LR exposure increased the contents of ROS, H_2_O_2_, and O_2−_ in the M10 and M30 groups and MDA (lipid peroxidation products) in the M30 group, indicating that oxidative stress was induced after MC-LR exposure in the liver of Nile tilapia (Table 2). 

### 2.2. Hepatic Histological Characteristics

As shown in Figure 1a, H&E staining results showed that the M10 group exhibited minor structural damages (e.g., blurred cellular boundaries and translocated nuclei) and higher lipid accumulation in the liver versus the control. However, the M30 group exhibited serious histological damage in the liver, as demonstrated by irregular hepatocyte arrangement, blurred cellular boundaries, and translocated, decreased, and atrophic nuclei, indicating hepatic damage induced by chronic MC-LR exposure. Meanwhile, MC-LR exposure resulted in increased hepatic cellular vacuolization in the Nile tilapia liver, demonstrated by significantly increased relative areas for liver vacuoles in the M30 group (*p* < 0.05) (Figure 1a,c), which corresponds with Oil Red O staining results of increased liver vacuoles observed in the M30 group (Figure 1b,d). 

Moreover, the M30 group exhibited significantly higher fluorescence intensity than the control (*p* < 0.05) (Figure 2a,c), indicating the accumulation of ROS in the liver of Nile tilapia after MC-LR exposure, which was in line with biochemical results (Table 2). Additionally, MC-LR exposure induced significantly higher apoptotic signals and the apoptotic rate in the M30 group versus the control (Figure 2b,d). 

As most determined endpoints were shown in the M30 group, liver samples of M30 and the control groups were used for further transcriptomic and metabolomic analysis.

### 2.3. Liver Transcriptomics

To clarify the potential molecular mechanism of liver damage induced by MC-LR exposure, transcriptomics was performed to screen differentially expressed genes (DEGs) between the M30 group and the control (Figure 3). PCA analysis showed a significant difference in the transcriptome profiles between the M30 group and the control (Figure 3a). Specifically, 3419 (1900 up- and 2419 down-regulated) genes were found between the M30 and the control groups (Figure 3b). GO enrichment analysis indicated that MC-LR exposure mainly affected the biological process and molecular function (Appendix A). For the biological process, DEGs were significantly enriched in different catabolic processes, metabolic processes, and biosynthetic processes after exposure to 30 μg/L MC-LR (Appendix A). For molecular function, DEGs were primarily related to enzyme activity and substance bindings (Appendix A). Then, DEGs were mapped onto KEGG pathways. The enrichment pathway in the M30 group mainly involved oxidative stress (peroxisome, chemical carcinogenesis-reactive oxygen species, and glutathione metabolism), bile metabolism (bile secretion and primary bile acid biosynthesis), lipid metabolism (e.g., biosynthesis of unsaturated fatty acids, fatty acid degradation, and PPAR signaling pathway), amino acid metabolism (e.g., histidine metabolism, arginine and proline metabolism, and glycine, serine, and threonine metabolism), apoptosis, and autophagy (Figure 3c). Lastly, a PPI network analysis was conducted to generate a network diagram, which was divided into six parts (oxidative stress, bile metabolism, lipid metabolism, amino acids metabolism, apoptosis, and autophagy) consisting of the forty-eight key DEGs in those pathways (Figure 3d). 

### 2.4. Liver Metabolomic Profiles

As shown in Figure 4a, principal component analysis (PCA) analysis showed a significant difference in the metabolic profiles between the M30 group and the control, indicating that chronic MC-LR exposure exerted significant effects on liver metabolism. Additionally, 940 (834 up- and 106 down-regulated) differential metabolites (DMs) were identified between the M30 group and the control (VIP value > 1 and *p*-value < 0.05) (Figure 4b and Appendix A). After mapping the DMs onto the KEGG pathways (Figure 4c), ten significantly enriched pathways were observed in the M30 group (*p* < 0.05), including glycine, serine and threonine metabolism, necroptosis, autophagy–animal, taurine and hypotaurine metabolism, primary bile acid biosynthesis, ascorbate and aldarate metabolism, histidine metabolism, d-amino acid metabolism, arginine biosynthesis, and lysine degradation (Figure 4c). A total of forty-two key metabolites in the ten key pathways were identified, such as taurocholic acid, glutathione, and d-Arabinono-1,4-lactone (Figure 4d).

### 2.5. The Expression Levels of Genes in the Liver

MC-LR exposure decreased hepatic cyp7a1 expression in the M30 group (*p* < 0.05) (Figure 5a). Meanwhile, MC-LR exposure markedly changed the expression levels of genes correlated with oxidative stress. Specifically, MC-LR exposure notably decreased the mRNA levels of *sod*, *cat*, and *gst* in the M30 group, as well as g6pd in the M10 and M30 groups, versus the control (*p* < 0.05) (Figure 5b). Furthermore, MC-LR exposure significantly lowered the mRNA levels of genes associated with lipid metabolism, as illustrated by the decreased expression of *cpt1a* and *acadm* in the M30 group and *acox3*, *acaa2*, and *hadhb* in the M10 and M30 groups, compared to the control (*p* < 0.05) (Figure 5c). Moreover, MC-LR exposure significantly changed the expression levels of genes related to apoptosis, as demonstrated by increased *caspase-3* expression in the M10 and M30 groups and declined expression of *bcl-2* and *xiap* in the M30 group versus the control, further confirming the occurrence of apoptosis in the liver of Nile tilapia chronically exposed to MC-LR (*p* < 0.05) (Figure 5d). In addition, compared to the control, the up-regulated expression of *atg4b*, *atg5*, and *atg12* was observed in the M30 group, suggesting the occurrence of autophagy after chronic MC-LR exposure (*p* < 0.05) (Figure 5d). 

### 2.6. Liver Bile Acid Contents

As shown in Figure 6, MC-LR exposure decreased the contents of liver TBA, CA, TCA, and TCDCA in the M30 group versus the control (*p* < 0.05). 

### 2.7. Integrated Metabolomic, Transcriptomic, and Biochemical Analysis Revealed the Mechanism of Hepatotoxicity Induced by Chronic MC-LR Exposure

To further investigate the hepatotoxicity mechanism caused by chronic MC-LR exposure, integrated metabolomic, transcriptomic, and biochemical analysis were used to identify several key pathways that induced the hepatotoxicity of MC-LR. The potential key pathways mainly included amino acid metabolism, lipid metabolism, redox regulation, primary bile acid biosynthesis, apoptosis, and autophagy (Figure 7).

## 3. Discussion

### 3.1. MC-LR Exposure Induced Hepatotoxicity

The liver is the primary targeted organ of MC-LR. A previous study reported that chronic exposure to environmental concentrations of MC-LR (10 μg/L) caused dysfunctions of mitochondria and disturbed lipid metabolism in the liver of zebrafish [1]. Consistent with the previous study, Nile tilapia exposed to 10 μg/L MC-LR exhibited minor structural damages (e.g., blurred cellular boundaries and translocated nuclei) and slight lipid accumulation. However, chronic exposure to 30 μg/L MC-LR induced serious histological damage (e.g., irregular hepatocyte arrangement, blurred cellular boundaries, translocated, decreased, and atrophic nuclei), severe lipid accumulation, over-accumulated ROS, and an increased apoptotic signal in the liver of Nile tilapia, which was in line with the enhanced activities of AST, ALT, and AKP, which were released into blood in the case of hepatic damage [9]. These results indicated that chronic exposure to higher environmental concentrations of MC-LR (30 μg/L) not only induced lipid accumulation but also resulted in severe hepatotoxicity due to increased oxidative stress and apoptosis. Thus, further investigation was needed to clarify the toxicity mechanism of MC-LR on the liver of fish at molecular levels.

### 3.2. MC-LR Exposure Caused Oxidative Stress by Reducing Antioxidant Status and Triggering Apoptosis and Autophagy

Oxidative stress is a major biological response process underlying environmental pollutant-induced toxicity. MC-LR exposure was widely reported to cause hepatic oxidative stress due to the reduced antioxidant enzyme activities in different experimental models, including mice [9], zebrafish [10], frogs [4], and common carp [3]. In agreement with previous studies, MC-LR exposure down-regulated the expression of antioxidant-related genes (e.g., *sod1*, *sod2*, *cat*, *prdx1*, *ephx2*, *gsta*, *gstk1*, *gstz1*, and *pex14*). SOD, CAT, PRDX1, and EPHX1 are key enzymes of antioxidant systems that are essential for scavenging free radicals in the organism’s body [11]. *Gsta*, *gstk1*, and *gstz1* encoded the GST that plays an important role in the antioxidant process via the conjugation of GSH with peroxides and free radicals [12]. Therefore, the decreased expression of antioxidant-related genes indicated that long-term MC-LR exposure induced oxidative stress due to ROS over-accumulated by inhibiting the antioxidant enzymes activities, which corresponds with the biochemical results (i.e., increased ROS content and decreased activity of SOD, CAT, GST, and GSH concentration). Notably, GSH was involved in the detoxification process by conjugating with numerous substrates, such as pharmaceuticals and environmental pollutants. The decreased GSH content in the liver may indicate an intense ongoing MC-LR detoxification process in Nile tilapia under MC-LR exposure [13]. Similarly, Jiang et al. (2011) reported that chronic exposure to MC-LR (10 μg/L) for 14d decreased CAT activity and GSH content, resulting in increased ROS accumulation and oxidative stress in the liver of common carp [3].

Oxidative stress from over-accumulated ROS induces oxidative damage to lipids, proteins, and nucleic acids, resulting in programmed cell death, including apoptosis and autophagy [14]. MC-LR is known to cause hepatotoxicity by inducing apoptosis and autophagy in hepatocytes [15]. In this study, tunnel staining results exhibited that apoptosis was significantly increased in Nile tilapia exposed to MC-LR. The results of transcriptomics and metabolomics also confirmed the apoptotic process induced by MC-LR. The enriched KEGG pathways in liver transcriptomic analysis included apoptosis and autophagy. MC-LR exposure lowered the expression of anti-apoptotic genes (e.g., *bcl2*, *bcl2l1*, *birc5*, and *xiap*) and enhanced the expression of apoptosis-promoting genes (*caspase-3*). Bcl-2 family proteins are crucial for inhibiting apoptosis [16]. XIAP, a key cellular inhibitor of apoptosis, acts as a crucial regulator of caspases [17]. Meanwhile, metabolic results showed enhanced contents of SM (D18:0/16:1 (9Z)) and arachidonic acid while decreasing taurine content after MC-LR exposure. Among them, SM (D18:0/16:1 (9Z)) was found to be a specific biomarker of apoptosis [18]. Arachidonic acid is known to induce the occurrence of oxidative stress and apoptosis. Taurine plays multiple roles in protecting cells from oxidative stress and apoptosis as an antioxidant [19]. The changes in these metabolites further confirmed that MC-LR induced an apoptotic process due to increased oxidative stress. Consistent with the results in the present study, MC-LR was reported to induce autophagy and apoptosis in the ovary cells of grass carp in vitro, as demonstrated by increased Bcl-2 and Bcl-2/Bax ratio [20]. MC-LR caused endoplasmic reticulum stress in the liver of zebrafish via the PERK, ATF6, and IRE1 pathways, which, in turn, resulted in apoptosis [15].

More importantly, the occurrence of autophagy induced by MC-LR is also tightly related to ROS-induced oxidative stress. Autophagy is a complex process of degrading dysfunctional cellular constituents inside cells via lysosomes under the regulation of autophagy-related genes (ATGs) [16]. Cathepsins in lysosomes are involved in different steps of autophagy [21]. After MC-LR exposure, the mRNA levels of autophagy-related genes (*atg4b*, *atg5*, *atg12*, *atg16l2*, and *dapk1*) and cathepsin-related genes (*ctsh*, *ctsk*, and *ctss*) were increased. DAPK1 plays key roles in regulating apoptosis and autophagy, whose phosphorylation activity induces programmed cell death [22]. Moreover, metabolic results displayed that MC-LR exposure enhanced the content of phosphatidylethanolamine (PE (18:4(6Z,9Z,12Z,15Z)/24:1(15Z)) and PE (22:6(4Z,7Z,10Z,13Z,16Z,19Z)/16:0)), which could promote the autophagic process, such as autophagosome elongation [23]. Thus, these results indicated that long-term MC-LR exposure may increase the autophagy in the liver of Nile tilapia. Overall, the transcriptional and metabolic results suggested that chronic MC-LR exposure caused apoptosis and autophagy.

### 3.3. MC-LR Exposure Disturbed Lipid Metabolism

MC-LR exposure was reported to induce a disturbance of lipid metabolism, which was related to numerous liver diseases, such as hepatitis and a fatty liver [4,24]. Biochemical results showed that chronic MC-LR exposure enhanced the serum contents of TG and TC of Nile tilapia, indicating dyslipidemia induced by MC-LR. Meanwhile, histological and biochemical results also showed that long-term MC-LR exposure induced lipid accumulation in the liver of Nile tilapia, as depicted by lipid vacuoles, lipid droplet accumulation, and increased TC and TG contents, particularly in the M30 group. Transcriptomic profiles revealed that chronic MC-LR exposure notably lowered the expression of fatty acids (FAs) and degradation-related genes (e.g., *cpt1a*, *acox3*, *acadl*, *acadm*, *ehhadh*, *acaa2*, *hadhb*, and *acat2*). These genes play key regulated roles in producing acetyl-CoA by the β-oxidation of FAs in the liver. Among them, CPT1A is a key rate-limiting enzyme that is responsible for conjugating acyl groups of fatty acid-CoA to carnitine, which was an important step before the β-oxidation of FAs in mitochondria [25]. These results showed that MC-LR exposure inhibited the β-oxidation of FAs in the liver of Nile tilapia, which was consistent with the decreased content of l-carnitine, which plays a vital role in facilitating the β-oxidation of FAs by transferring acyl-CoA into mitochondria. Moreover, chronic MC-LR exposure also up-regulated the mRNA levels of FA synthesis-related genes, such as *acot1*. ACOT1 is responsible for converting acyl-CoAs to FAs [26]. Therefore, these findings suggested that MC-LR exposure increased lipid deposition in the Nile tilapia liver, mainly by down-regulating lipolytic genes. Similarly, Zhang et al. (2020) reported that long-term MC-LR exposure resulted in liver lipid accumulation by up-regulating the expression of lipogenic genes (e.g., *fasn*, *srebf1*, *srebf2*, and *acaca*) and down-regulating lipolytic genes (e.g., *cpt1a*, *atgl*, and *hsla*) in male zebrafish [5]. Chronic dietary exposure to MC-LR led to liver lipid accumulation due to the up-regulated lipid synthesis and inhibited FA β-oxidation in mice [2]. Altogether, the observed results suggested that chronic exposure to MC-LR caused hepatic lipid accumulation by limiting FA β-oxidation and the up-regulation of lipid synthesis in Nile tilapia.

### 3.4. MC-LR Exposure Disturbed Amino Acid Metabolism

Moreover, transcriptomic and metabolomic results demonstrated that MC-LR exposure affected the hepatic amino acid metabolism of Nile tilapia. Specifically, MC-LR exposure decreased the contents of glutamate, l-histidine, and citruline, while increasing histamine content. Glutamate is an important precursor of GSH, which plays a vital role in mitigating the toxicity of MC-LR by binding it to form a soluble substance or scavenging over-accumulated free radicals induced by it [27]. The decreased glutamate concentration could be ascribed to the continued depletion of GSH [28]. Moreover, histamine is released when tissues are damaged or when inflammation and allergic reactions occur [29]. The increased histamine content indicated tissue damage in the liver of Nile tilapia, which was in line with histological results. The increased histamine content in metabolites may be closely associated with the up-regulated expression of *hdc* (histidine decarboxylase), which is a key enzyme that catalyzes histidine decarboxylation to produce histamine [30]. Additionally, histamine could be oxidized and decomposed into biologically active molecules by amine oxidase (AO), including aldehyde, NH_3_, and H_2_O_2_ [31]. The down-regulated expression of *ao* may also be partly responsible for the increased concentration of histamine. Altogether, these results demonstrated that chronic MC-LR exposure disturbed amino acid metabolism, which may indirectly cause tissue damage in the liver of Nile tilapia. 

### 3.5. MC-LR Exposure Inhibited BA Synthesis

BAs are crucial for maintaining liver homeostasis, and decreased BAs may cause different liver diseases (e.g., fatty liver and steatohepatitis) [32]. Cholesterol is the main precursor substance for the synthesis of hepatic primary bile acids, and thus, BA synthesis is important for cholesterol homeostasis [6]. In the present study, we show the transcription of BA synthesis-related genes (e.g., *cyp7a1*, *cyp46a1*, *ch25h*, *hsd3b7*, *amacr*, and *hsd17b4*). Cyp7a1 encodes the key rate-limiting enzyme that catalyzes cholesterol to form 7α-hydroxycholesterol, which is the first step of the neutral pathway for synthesizing chenodeoxycholate and cholate [25]. *Amacr* and *hsd17b4* also encode the key enzymes in the neutral pathway of BA synthesis [33]. Moreover, *cyp46a1* and *ch25h* encode the first rate-limiting enzymes in the 24-hydroxylase pathway and 25-hydroxylase pathway, respectively [33]. Thus, the down-regulated expression of these genes indicated that MC-LR exposure inhibited hepatic primary BA synthesis, which was consistent with the targeted metabolomic results that BA concentrations were decreased in Nile tilapia after MC-LR exposure. Additionally, it is notable that MC-LR exposure decreased the expression of the BA secretion-related gene (*abcc2*) while increasing the expression of the BA recycling-related gene (*slc22a7*). ABCC2 belongs to the subfamily of the ABC transporter family that is involved in mediating bile acid transport from hepatocytes into bile canaliculus [34]. *Slc22a7* encoded bile acid transporters, such as OATP2, which promote the BA reabsorption from the intestine [35]. The decreased *abcc2* expression and increased *slc22a7* expression indicated the deficiency of BAs in the Nile tilapia liver after MC-LR exposure, which may be attributed to the down-regulation of BA synthesis. Therefore, the decreased levels of BA synthesis-related genes suggested that long-term MC-LR exposure suppressed the BA synthesis in the liver of Nile tilapia. In line with these results, a previous study reported that acute exposure to MC-LR lowered the relative expression level of *cyp7a1* in the liver of Chinese mitten crab (*Hypophthalmicthys molitrix*) [36]. Another study on shrimp showed that MC-LR exposure decreased the intestinal levels of taurocholate and taurodeoxycholic acid, affecting the bile acid metabolism of shrimp [37]. Moreover, consistent with the transcriptomic results, metabolic results showed that BA synthesis raw material (taurine) and primary BA (cholic acid and taurocholic acid) levels in the Nile tilapia liver decreased after exposure to MC-LR. BAs play key roles in regulating the lipid metabolism in fish by modulating the FXR receptor, and its downstream targets are responsible for controlling de novo lipogenesis, fatty acid oxidative metabolism, and triglyceride hydrolysis in the liver [25]. Therefore, these results confirmed that impeded BA synthesis was an important cause of hepatotoxicity after MC-LR exposure.

## 4. Conclusions

This study showed that long-term exposure to higher environmental concentrations of MC-LR (30 μg/L) induced serious histological damage and liver lipid accumulation in Nile tilapia. The integrated metabolomic, transcriptomic, and biochemical analysis uncovered the important biomolecules and biological pathways related to the potential mechanism of MC-LR-induced hepatotoxicity. Taken together, chronic MC-LR exposure could induce hepatotoxicity due to oxidative stress, apoptosis, autophagy, and lipid accumulation. Moreover, chronic MC-LR exposure decreased primary bile acid biosynthesis, which may increase lipid accumulation due to inhibited lipid decomposition in the liver of Nile tilapia. Therefore, this research could shed new light on the environmental health risks of MC-LR to aquatic organisms.

## 5. Materials and Methods

### 5.1. Fish Culture and Experimental Design

Juvenile Nile tilapia were purchased from a local aquaculture farm located in Guangzhou and acclimated in a plastic tank (400 × 200 × 50 cm) for 3 weeks for acclimatization. During this period, fish were fed twice daily to satiation using commercial feed shown in Appendix A. A total of 300 healthy juvenile Nile tilapia (initial weight 4.57 ± 0.36 g) were randomly assigned to 15 tanks (100 × 50 × 25 cm) with 3 replicates per group. The experiment included five groups: C (control, 0 μg/L MC-LR), M1 (1 μg/L MC-LR), M3 (3 μg/L MC-LR), M10 (10 μg/L MC-LR), and M30 (30 μg/L MC-LR). During the experimental period, Nile tilapia were fed to satiation twice daily (9.00 and 18.00) for 60 days. The feces were removed using a siphon every day. Water conditions in all aquaria were as follows: water temperature 26–29 °C, dissolved oxygen 7.0–7.8 mg/L, and pH 7.2–7.8. 

The lyophilized solids of MC-LR (5 mg, purity > 95% by HPLC) were purchased from Taiwan Algae Science Inc (Taiwan, China). A solution of MC-LR (0.5 mg/mL) was prepared by adding 10 mL of ultrapure water. To keep the concentration of MC-LR stable during the whole exposed experiment, the MC-LR concentration of exposed water was determined using commercial ELISA kits (Mlbio, Shanghai, China) twice daily (once in 12 h) to adjust MC-LR concentration, and half of the exposed water was substituted with clean water containing different corresponding concentrations of MC-LR once a week [38].

### 5.2. Fish Sampling

At the end of the 60-day exposed experiment, nine Nile tilapia in each group were selected (three replicates per tank) and anesthetized with MS-222 solution (20 mg/L) in a plastic tank (50 × 40 × 25 cm). Blood samples of the nine fish were collected using 1 mL syringes and put in 2 mL centrifuge tubes for standing for 4 h at 4 °C to obtain serum after centrifuging at 3500 rpm (15 min, 4 °C). Liver tissues of the nine fish were collected for the subsequent analysis of biochemical parameters and gene expression analysis. Meanwhile, the liver tissues of the six fish were collected and divided into five parts. Three parts of them were prepared for transcriptomics, metabolomics, and dihydroethidium (DHE) staining. Three parts of them (2 mm^3^) were fixed using 4% paraformaldehyde for Oil Red O staining, hematoxylin–eosin (H&E) staining, and TUNEL staining. The samples (serum and liver) described above, apart from the liver samples fixed in paraformaldehyde, were frozen with liquid nitrogen and stored at −80 °C during sampling.

### 5.3. Histological Examination, DHE Staining, and TUNEL Staining

#### 5.3.1. H&E Staining

H&E staining was conducted as described in a previous study with minor modifications [39]. Briefly, the fixed hepatic tissues were dehydrated in graded ethanol series, a mixture of ethanol and xylene (1:1), xylene, and paraffin. Next, the tissues were embedded and sliced into thin sections (4 μm). Then, they were dehydrated in xylene and ethanol and washed with clean distilled water. Subsequently, the sections were stained with hematoxylin–eosin dye solution (B1002, Baiqiandu Biotechnology, Wuhan, China). The sections were observed with an Olympus BX41 microscope (Olympus Cor., Tokyo, Japan).

#### 5.3.2. Oil Red O Staining

Liver sections were stained with neutral Oil Red O (Sigma-Aldrich) to visualize the hepatic fatty droplets [40]. Briefly, the fixed liver tissues were embedded and rapidly frozen using isopentane cooled in liquid nitrogen. The tissues were then sliced into thin sections (5–10 μm) using a cryostat, which were washed with clean distilled water. Subsequently, after staining with Oil Red O, slides were destained with 60% isopropanol and washed with clean distilled water, and the slides were counterstained with Mayer’s hematoxylin. After the staining and rinsing steps, the sections were observed with an Olympus BX41 microscope (Olympus Corporation., Tokyo, Japan).

#### 5.3.3. Dihydroethidium (DHE) Staining

DHE staining was employed to detect ROS in liver tissues [41,42]. Briefly, the liver tissues were embedded in a medium (Sakura, SAKURA 4583, Tokyo, Japan). Then, after cutting into thin sections (5–10 μm), the sections were dyed with DHE solution (Sigma, D7008, St. Louis, MO, USA). Lastly, the sections were observed with a 3D Histech fluorescence scanner (Pannoramic MIDI, Ltd., Budapest, Hungary). 

#### 5.3.4. TUNEL Staining

TUNEL assays were conducted to detect the initiation of apoptosis in the liver [43]. Briefly, fixed hepatic tissues were cut into thin sections (5 μm), which were baked in a 65 °C oven for 2 h. The slices were then dehydrated with xylene and graded ethanol and washed with clean distilled water thrice. Next, the sections were incubated at 37 °C for 30 min with proteinase K and washed in PBS for 5 min three times for repairing and permeability. After adding 50 μL reaction solution of the tunnel (Roche, Basel, Switzerland) in the sections, avoid light incubation was conducted for 2 h at 37 °C. Then, after washing 3 times for 5 min each using PBS (PH = 7.4), the slices were stained using DAPI solution for 10 min. Lastly, the slices were sealed and observed using a microscope (Eclipse Ti-SR, Nikon, Tokyo, Japan). 

### 5.4. Biochemical Parameter Determination

Commercial kits from Jiancheng biotech (Nanjing, China) were used to determine serum biochemical parameters, including alkaline phosphatase (AKP), low-density lipoprotein cholesterol (LDL-C), triglycerides (TGs), alanine aminotransferase (ALT), high-density lipoprotein cholesterol (HDL-C), total cholesterol (TC), and aspartate aminotransferase (AST). Additionally, lipid metabolism parameters (TG, TC, LDL-C, and HDL-C) of the liver were also measured using commercial kits from Jiancheng biotech (Nanjing, China). Moreover, commercial ELISA kits were used to determine liver total bile acid (TBA) and oxidative stress parameters, including malondialdehyde (MDA), superoxide radicals (O_2_^−^), reactive oxygen species (ROS), hydrogen peroxide (H_2_O_2_), and hydroxyl-free radicals (OH^−^).

### 5.5. Transcriptomic Analysis

#### 5.5.1. Total Extraction of RNA and Sequencing

The total hepatic RNA was extracted using TRIzol (Invitrogen, CA, USA). Then, Nanodrop2000 was used to determine the concentration and purity of the RNA. The RIN values of the RNA were determined using a bioanalyzer (Agilent2100, Santa Clara, CA, USA). The library construction of RNA was built according to the following conditions: RNA concentration ≥ 35 ng/μL, total amount of RNA ≥ 1 ug, OD 260/230 ≥ 1, and OD 260/280 ≥ 1.8. Reverse transcription and sequencing were conducted with the Illumina Novaseq 6000-sequencing platform (Illumina, San Diego, CA, USA). 

#### 5.5.2. Transcript Assembly and Annotation

After the sequence comparative analysis using TopHat2 (http://tophat.cbcb.umd.edu/, accessed on 20 January 2024), the assembly and splice of mapped reads were performed using Cufflinks (http://coletrapnelllab.github.io/cufflinks/, accessed on 20 January 2024) to obtain transcripts with no annotated information, which were used for further functional annotation.

#### 5.5.3. Analysis of Differentially Expressed Genes

The results of the alignment to the genome and genome annotation files from RSEM were used to obtain read counts of the transcripts [44], which were used for FPKM or TPM transformation to measure the gene expression levels. The differentially expressed genes (DEGs) were screened out ((|log2(FC)| > 1 and *p* < 0.05)). The functional enrichment analysis of DEGs (GO enrichment and KEGG pathway enrichment) was performed using Goatools and KOBAS, respectively. A network diagram was generated via protein–protein interaction (PPI) network analysis to clarify the relations of key DEGs, according to our previous study [45]. The raw data were deposited in the NCBI Sequence Read Archive (SRA) with accession number PRJNA1066979. The data was accessed on 20 January 2024.

### 5.6. Metabolomic Analysis

Untargeted metabolomics was performed to extract and analyze metabolites using LC-MS [41]. Briefly, the metabolites of liver tissues (n = 6) were extracted in an extracting solution (methanol/water = 4:1). The metabolic profile of the extract was analyzed using a UHPLC-Q exactive system (Thermo Fisher Scientific, CA, USA) coupled to an ACQUITY UPLC HSS T3 column.

Raw data are preprocessed to reduce the impact of variation in the data that are not relevant to the purpose of this study on data analysis, including the filtering and padding of missing values, data normalization, QC (quality control) validation, and data conversion. Then, the raw data were analyzed by ProgenesisQI (WatersCorporation, Milford, MA, USA) to obtain quantitative information on metabolites combined with quality error (<10 ppm), such as retention time and peak intensity. The Majorbio Cloud Platform was used for further analysis of the preprocessed data, including PCA and KEGG pathway analysis.

Targeted metabolomics was employed to determine the contents of liver bile acids [46,47,48], including cholic acid (CA), taurocholate acid (TCA), taurochenodeoxycholic acid (TCDCA), chenodeoxycholic acid (CDCA), and glycocholic acid (GCA). Briefly, liver tissues were ground in 400 μL of methanol at 55 Hz for 1 min twice to obtain homogenate, which was centrifuged to obtain the supernatant [46]. The supernatant was used for bile acid analysis using UPLC-MS/MS (AB6500 Plus, SCIEX, Framingham, MA, USA) [49].

### 5.7. qPCR Analysis

The hepatic total RNA was extracted with TRIzol reagent (Invitrogen, Carlsbad, CA, USA). Then, RNA was reversed to cDNA using a PrimeScript RT reagent kit (TaKaRa, Dalian, China). The following programs were used to perform qRT-PCR: 95 °C for 30 s, 60 cycles of 95 °C for 5 s and 60 °C for 30 s, 95 °C for 5 s, 60 °C for 60 s, 95 °C for 1 s, and 50 °C for 30 s. The 2^−ΔΔCt^ method was used to calculate gene expression with β-actin as a reference gene. The primers of the genes are listed in Appendix A.

### 5.8. Statistic Analysis

Student’s *t*-test was employed to analyze the difference in liver bile acid contents between the M30 group and the control. One-way ANOVA followed by Duncan’s test were used to analyze other data, including the hepatopancreas somatic index, biochemical results of the serum and liver, and gene expression levels, using SPSS 23 (SPSS Inc., Chicago, IL, USA).

## Figures and Tables

**Figure 1 toxins-16-00149-f001:**
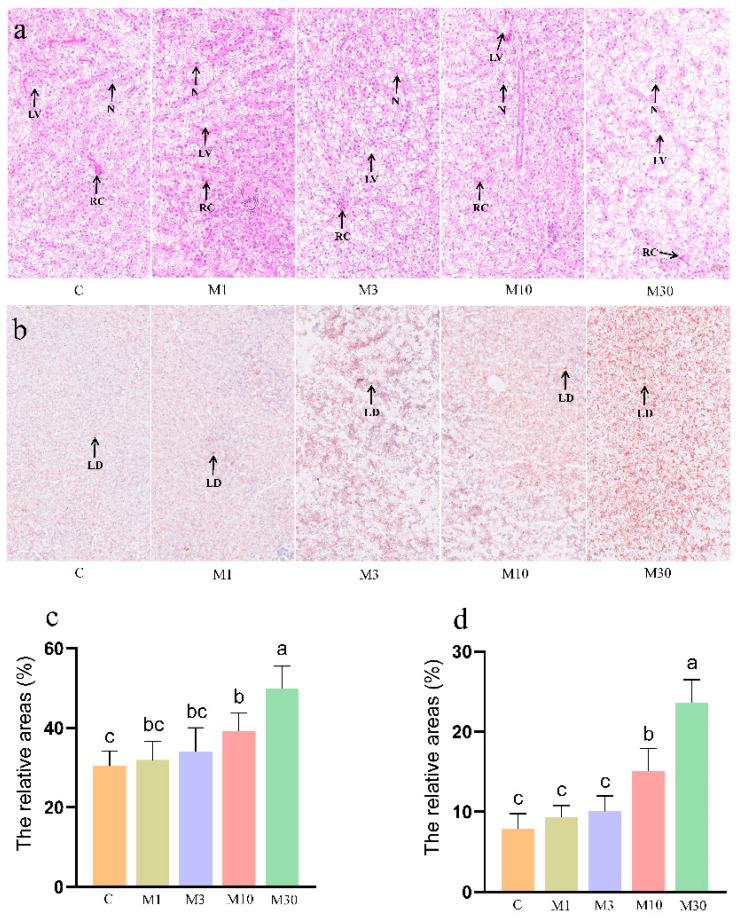
Histological characteristics of the Nile tilapia liver. (**a**) Hematoxylin–eosin (H&E) staining. Lipid vacuole (LV), nucleus (N), and red cell (RC). (**b**) Oil Red O staining. Lipid droplets (LDs). (**c**,**d**) The relative areas for liver vacuoles in the H&E stain and lipid droplets in the Oil Red O stain. Values are reported as mean ± standard deviation (SD) and normalized to the percentage of field area. Values of the same parameters with different letters were significantly different in concentration groups (*p* < 0.05, n = 6). Data are presented as means ± SD. C, control, 0 μg/L. MC-LR; M1, 1 μg/L MC-LR; M3, 3 μg/L MC-LR; M10, 10 μg/L MC-LR; M30, 30 μg/L MC-LR.

**Figure 2 toxins-16-00149-f002:**
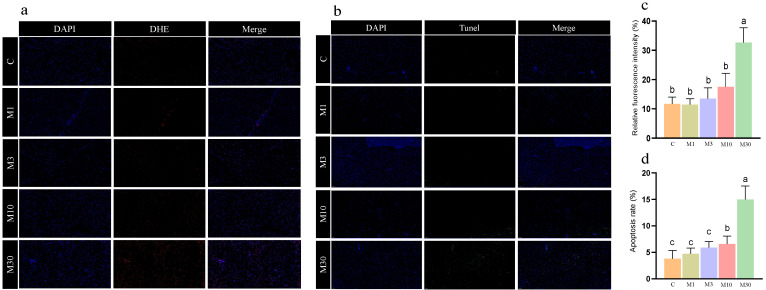
The results of DHE staining and TUNEL staining of the Nile tilapia liver. (**a**) Dihydroethidium (DHE) staining (20×). Representative DAPI and DHE staining of sections of the Nile tilapia liver. (**b**) Terminal–deoxynucleotidyl transferase-mediated nick end labeling (TUNEL) staining (20×). Representative DAPI and TUNEL staining of sections of the Nile tilapia liver. The images were analyzed using Image-Pro Plus 4.1 software. Positive apoptotic nuclei and normal nuclei are shown in green and blue, respectively. (**c**) Quantitative analysis of the mean fluorescence intensity of DHE using ImageJ software. (**d**) Apoptosis rate. Apoptotic index: the number of apoptotic nuclei/the number of observed nuclei × 100%. C, control, 0 μg/L MC-LR; M1, 1 μg/L MC-LR; M3, 3 μg/L MC-LR; M10, 10 μg/L MC-LR; M30, 30 μg/L MC-LR. Values of the same parameters with different letters were significantly different in concentration groups (*p* < 0.05, n = 6). Data are presented as means ± SD.

**Figure 3 toxins-16-00149-f003:**
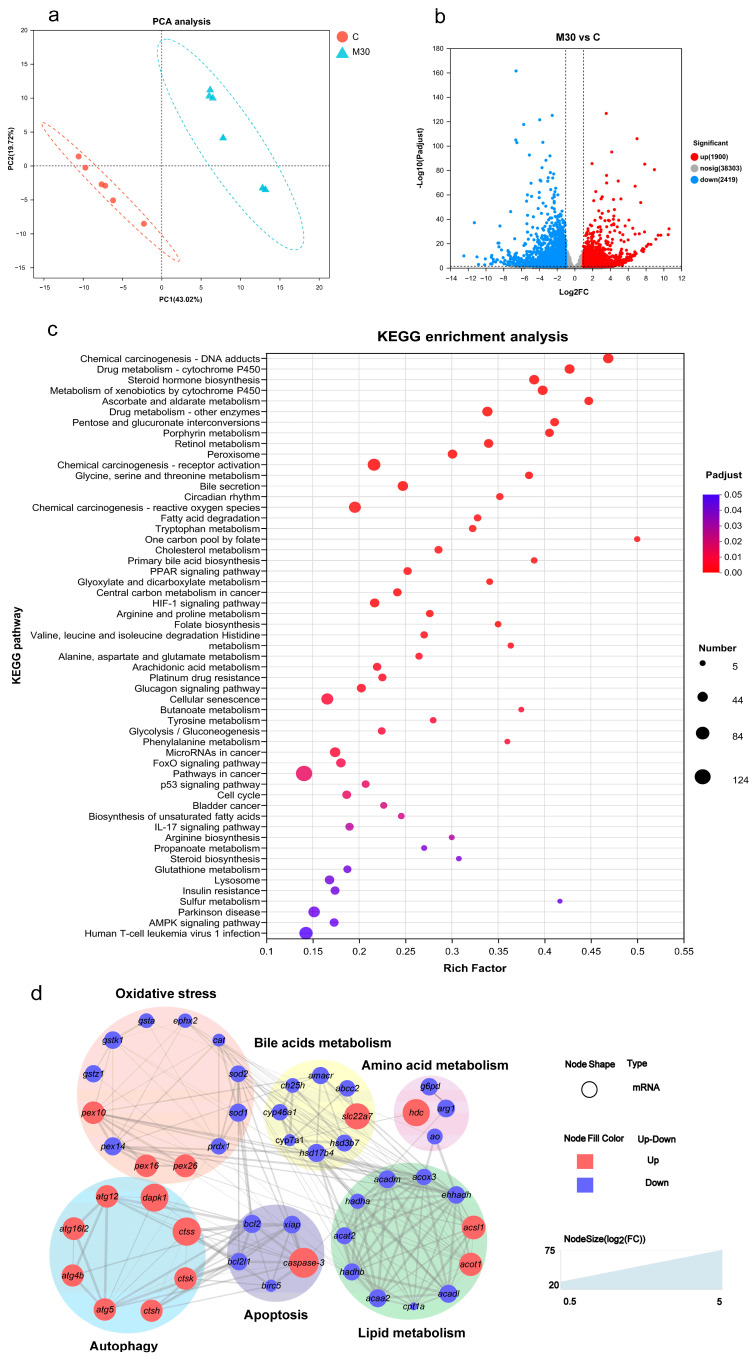
Effects of chronic MC-LR exposure on the transcriptomic features of the Nile tilapia liver. (**a**) PCA analysis for the transcriptomic profiles. (**b**) Volcano plot of the liver of differentially expressed genes (DEGs). (**c**) Bubble diagram of the KEGG enrichment analysis of DEGs. (**d**) The protein–protein interaction network in the grass carp of genes between the C and M30 groups. C, control, 0 μg/L MC-LR; M30, 30 μg/L MC-LR. The larger the size of the node shape, the greater the FC of mRNAs. A combined score > 0.4 was considered a statistically significant interaction. *cat*, catalase; *sod1*, superoxide dismutase 1; *sod2*, superoxide dismutase 2; *prdx1*, peroxiredoxin-1; *ephx2*, epoxide hydrolase 2; *gsta*, glutathione S-transferase; *gstk1*, glutathione S-transferase kappa 1; *gstz1*, glutathione S-transferase zeta 1; *pex10*, peroxisomal biogenesis factor 10; *pex14*, peroxisomal biogenesis factor 14; *pex16*, peroxisomal biogenesis factor 16; *pex26*, peroxisomal biogenesis factor 26. *ch25h*, cholesterol 25-hydroxylase; *cyp46a1*, cholesterol 24-hydroxylase; *hsd3b7*, hydroxy-delta-5-steroid dehydrogenase, 3 beta- and steroid delta-isomerase 7; *cyp7a1*, cholesterol 7-alpha-monooxygenase; amacr, alpha-methylacyl-CoA racemase; hsd17b4, hydroxysteroid 17-beta dehydrogenase 4; *abcc2*, canalicular multispecific organic anion transporter 1; *slc22a7*, solute carrier family 22 member 7. *cpt1a*, carnitine O-palmitoyltransferase 1; *acox3*, acyl-CoA oxidase 3, pristanoyl; *acadl*, acyl-CoA dehydrogenase long chain; *acadm*, acyl-CoA dehydrogenase medium chain; *ehhadh*, enoyl-CoA hydratase and 3-hydroxyacyl CoA dehydrogenase; *hadha*, trifunctional enzyme subunit alpha, mitochondrial; *acaa2*, acetyl-CoA acyltransferase 2; *acat2*, acetyl-CoA acetyltransferase 2; *acot1*, acyl-coenzyme A thioesterase 1; *acsl1*, long-chain fatty-acid-CoA ligase 1; *hadhb*, hydroxyacyl-CoA dehydrogenase trifunctional multienzyme complex subunit beta; *bcl2*, B-cell lymphoma-2; *bcl2l1*, BCL2 like 1; *caspase-3*, cysteinyl aspartate-specific proteinase 3; *birc5*, baculoviral IAP repeat-containing protein 5; *xiap*, X-linked inhibitor of apoptosis; *atg12*, autophagy-related 12; *atg16l2*, autophagy-related 16 like 2; *atg4b*, autophagy-related 4B cysteine peptidase; *atg5*, autophagy-related 5; *dapk1*, death-associated protein kinase 1; *ctsh*, cathepsin H; *ctsk*, cathepsin K; *ctss*, cathepsin S; *hdc*, histidine decarboxylase; *g6pd*, glucose-6-phosphate 1-dehydrogenase; *arg1*, arginase-1; *ao*, amine oxidase.

**Figure 4 toxins-16-00149-f004:**
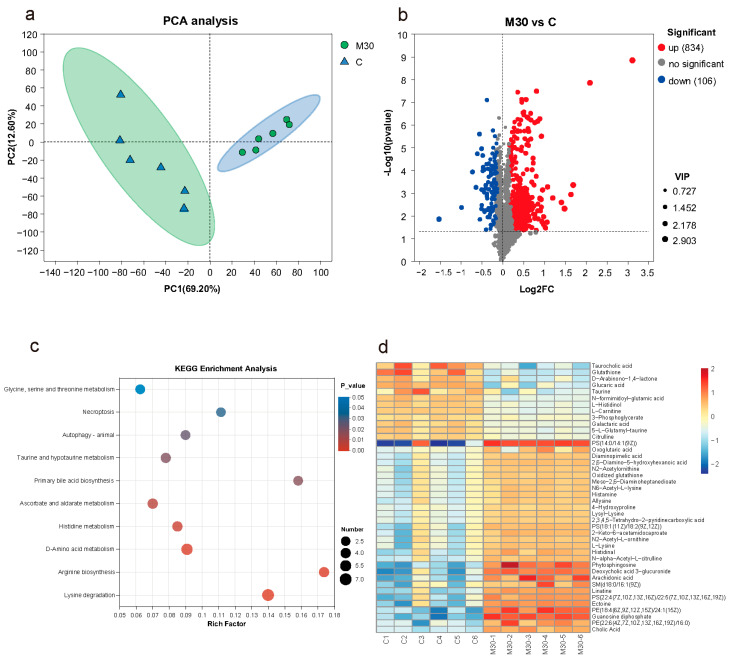
Effects of chronic MC-LR exposure on the metabolomic features of the Nile tilapia liver. (**a**) PCA analysis of the metabolomics profiles. (**b**) Volcano plot of liver differential metabolites. (**c**) Bubble diagram of the KEGG enrichment analysis of liver differential metabolites. (**d**) Heatmap visualization of key differential metabolites; Each row of the heatmap was labeled with tentative metabolite names, and the colors refer to the relative levels of these compounds from high (red) to low (blue).

**Figure 5 toxins-16-00149-f005:**
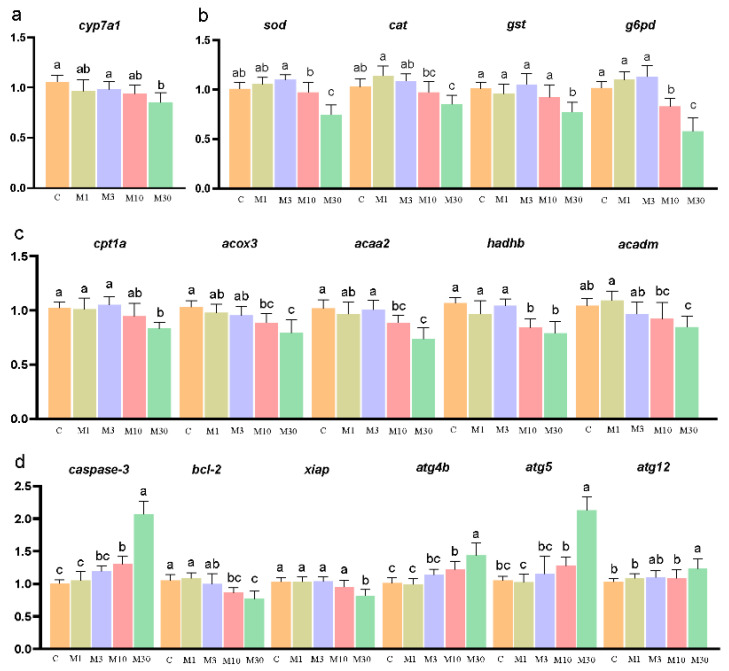
Effects of chronic MC-LR exposure on the expression of key genes in the Nile tilapia liver. (**a**) Bile acid metabolism-related genes; (**b**) oxidative stress-related genes; (**c**) lipid metabolism-related genes; (**d**) apoptosis and autophagy-related genes. Data are presented as means ± SD. Values of the same parameters with different letters were significantly different in concentration groups (*p* < 0.05, n = 9). Data are presented as means ± SD. *cyp7a1*, cholesterol 7-alpha-monooxygenase; *cpt1a*, carnitine O-palmitoyltransferase 1; *acox3*, acyl-CoA oxidase 3, pristanoyl; *acaa2*, cetyl-CoA acyltransferase 2; *hadhb*, hydroxyacyl-CoA dehydrogenase trifunctional multienzyme complex subunit beta; *acadm*, acyl-CoA dehydrogenase medium chain; *sod*, superoxide dismutase; *cat*, catalase; *gst*, glutathione S-transferase; *g6pd*, glucose-6-phosphate 1-dehydrogenase; *caspase-3*, cysteinyl aspartate specific proteinase 3; *bcl2*, B-cell lymphoma-2; *xia*p, X-linked inhibitor of apoptosis; *atg4b*, autophagy-related 4B cysteine peptidase; *atg5*, autophagy-related 5; *atg12*, autophagy-related 12. C, control, 0 μg/L MC-LR; M1, 1 μg/L MC-LR; M3, 3 μg/L MC-LR; M10, 10 μg/L MC-LR; M30, 30 μg/L MC-LR.

**Figure 6 toxins-16-00149-f006:**
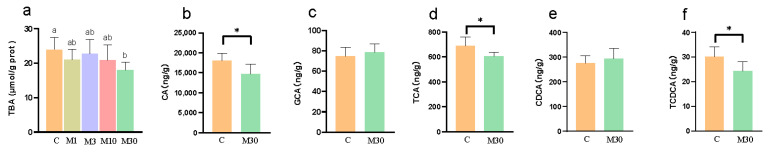
Effects of chronic MC-LR exposure on the contents of bile acids in the Nile tilapia liver. (**a**) TBAs, total bile acids, and values of the same parameters with different letters were significantly different in concentration groups (*p* < 0.05, n = 9); (**b**) CA, cholic acid; (**c**) GCA, glycocholic acid; (**d**) TCA, taurocholate acid; (**e**) CDCA, chenodeoxycholic acid; (**f**) TCDCA, taurochenodeoxycholic acid, and values of the same parameters with an asterisk were significantly different (*p* < 0.05, n = 6). Data are presented as means ± SD. C, control, 0 μg/L MC-LR; M1, 1 μg/L MC-LR; M3, 3 μg/L MC-LR; M10, 10 μg/L MC-LR; M30, 30 μg/L MC-LR.

**Figure 7 toxins-16-00149-f007:**
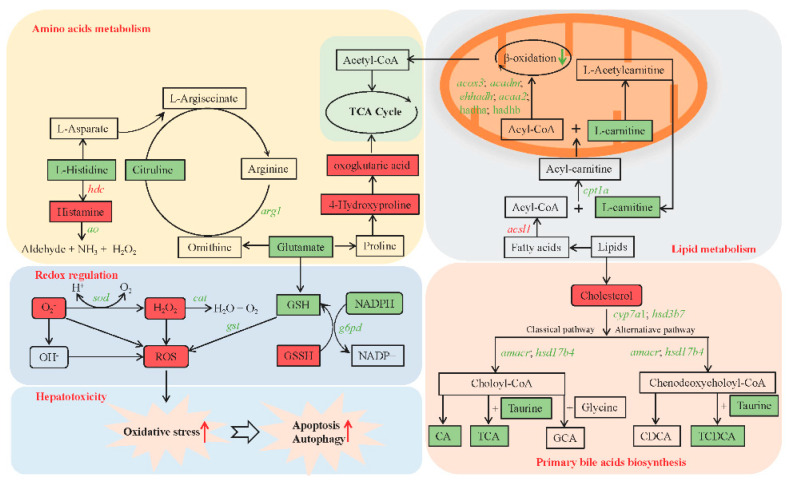
Integrated metabolomic and transcriptomic analysis of key pathways in the Nile tilapia liver in response to chronic MC-LR exposure. The rectangle represents key metabolites. Inclined letters represent key genes. Red and green represent higher or lower levels, respectively, in the M30 group compared to the control group. C, control, 0 μg/L MC-LR; M30, 30 μg/L MC-LR. CA, cholic acid; GCA, glycocholic acid; TCA, taurocholate acid; CDCA, chenodeoxycholic acid; TCDCA, taurochenodeoxycholic acid. *cpt1a*, carnitine O-palmitoyltransferase 1; *acox3*, acyl-CoA oxidase 3, pristanoyl; *acadm*, acyl-CoA dehydrogenase medium chain; *ehhadh*, enoyl-CoA hydratase and 3-hydroxyacyl CoA dehydrogenase; *acaa2*, cetyl-CoA acyltransferase 2; *hadha*, trifunctional enzyme subunit alpha, mitochondrial; *hadhb*, hydroxyacyl-CoA dehydrogenase trifunctional multienzyme complex subunit beta; *sod*, superoxide dismutase; *cat*, catalase; *gst*, glutathione S-transferase; *g6pd*, glucose-6-phosphate 1-dehydrogenase; *arg1*, arginase-1; *hdc*, histidine decarboxylase; *ao*, amine oxidase; *cyp7a1*, cholesterol 7-alpha-monooxygenase; hsd3b7, hydroxy-delta-5-steroid dehydrogenase, 3 beta- and steroid delta-isomerase 7; amacr, alpha-methylacyl-CoA racemase; hsd17b4, hydroxysteroid 17-beta dehydrogenase 4.

**Table 1 toxins-16-00149-t001:** The effects of chronic MC-LR exposure on the serum biochemical parameters of Nile tilapia.

	C	M1	M3	M10	M30
TG (mmol/L)	5.2 ± 0.40 ^c^	5.62 ± 0.95 ^bc^	5.59 ± 0.75 ^bc^	6.82 ± 1.48 ^ab^	7.15 ± 1.40 ^a^
TC (mmol/L)	3.86 ± 0.44 ^b^	4.12 ± 0.83 ^ab^	4.31 ± 0.77 ^ab^	5.03 ± 1.22 ^ab^	5.43 ± 1.16 ^a^
LDL-C (mmol/L)	3.32 ± 0.38 ^b^	3.45 ± 0.48 ^b^	3.91 ± 0.78 ^ab^	4.11 ± 0.82 ^ab^	4.73 ± 0.81 ^a^
HDL-C (mmol/L)	2.91 ± 0.43 ^c^	3.15 ± 0.72 ^c^	3.52 ± 0.65 ^bc^	4.10 ± 0.42 ^ab^	4.59 ± 0.92 ^a^
ALT (U/L)	21.38 ± 2.86 ^c^	24.39 ± 4.12 ^bc^	27.33 ± 6.03 ^bc^	28.42 ± 5.19 ^ab^	33.15 ± 4.81 ^a^
AST (U/L)	29.90 ± 4.70 ^b^	31.07 ± 4.64 ^b^	34.51 ± 4.74 ^ab^	36.02 ± 4.85 ^ab^	39.57 ± 5.17 ^a^
AKP (U/L)	133.98 ± 11.41 ^b^	143.00 ± 17.01 ^ab^	140.90 ± 13.15 ^ab^	151.90 ± 15.21 ^ab^	160.40 ± 13.79 ^a^

Note: C, control, 0 μg/L MC-LR; M1, 1 μg/L MC-LR; M3, 3 μg/L MC-LR; M10, 10 μg/L MC-LR; M30, 30 μg/L MC-LR. TGs, triglycerides; TC, total cholesterol; LDL-C, low-density lipoprotein cholesterol; HDL-C, high-density lipoprotein cholesterol; ALT, alanine aminotransferase; AST, aspartate aminotransferase; AKP, alkaline phosphatase. Values of the same parameters with different letters were significantly different in concentration groups (*p* < 0.05, n = 9). Data are presented as means ± SD.

**Table 2 toxins-16-00149-t002:** The effects of chronic MC-LR exposure on the liver biochemical parameters of Nile tilapia.

	C	M1	M3	M10	M30
Lipid metabolism parameters					
TG (mmol/L)	3.02 ± 0.37 ^c^	3.44 ± 0.62 ^c^	3.76 ± 0.62 ^c^	4.94 ± 0.47 ^b^	5.77 ± 0.68 ^a^
TC (mmol/L)	1.46 ± 0.19 ^b^	1.55 ± 0.14 ^b^	1.65 ± 0.28 ^b^	1.71 ± 0.21 ^ab^	1.97 ± 0.13 ^a^
LDL-C (mmol/L)	2.71 ± 0.45 ^c^	2.88 ± 0.43 ^c^	2.80 ± 0.32 ^c^	3.58 ± 0.41 ^b^	4.53 ± 0.55 ^a^
HDL-C (mmol/L)	1.68 ± 0.22 ^b^	1.77 ± 0.22 ^b^	1.79 ± 0.23 ^b^	1.92 ± 0.36 ^b^	2.44 ± 0.55 ^a^
Oxidative stress parameters					
MDA (nmol/mgprot)	25.15 ± 2.61 ^b^	25.27 ± 3.93 ^b^	28.08 ± 4.78 ^ab^	29.73 ± 3.51 ^ab^	31.69 ± 4.37 ^a^
ROS (U/g)	2533.54 ± 373.85 ^b^	2760.41 ± 599.13 ^ab^	2759.42 ± 374.84 ^ab^	3350.73 ± 440.91 ^a^	3201.53 ± 406.28 ^a^
H_2_O_2_ (μg/g)	16.71 ± 1.71 ^c^	17.42 ± 3.62 ^bc^	18.32 ± 3.15 ^bc^	21.51 ± 3.70 ^ab^	23.46 ± 3.24 ^a^
O_2_^-^ (μg/g)	84.67 ± 7.73 ^c^	91.56 ± 16.30 ^bc^	95.06 ± 10.83 ^bc^	106.44 ± 10.45 ^ab^	115.29 ± 13.03 ^a^
OH^-^ (μg/g)	53.47 ± 5.73	57.83 ± 9.71	54.97 ± 9.45	64.83 ± 9.81	56.76 ± 7.12

Note: C, control, 0 μg/L MC-LR; M1, 1 μg/L MC-LR; M3, 3 μg/L MC-LR; M10, 10 μg/L MC-LR; M30, 30 μg/L MC-LR. TGs, triglycerides; TC, total cholesterol; LDL-C, low-density lipoprotein cholesterol; HDL-C, high-density lipoprotein cholesterol; MDA, malondialdehyde; ROS, reactive oxygen species; H_2_O_2_, hydrogen peroxide; O_2−_, superoxide radicals; OH_−_, hydroxyl-free radical. Values of the same parameters with different letters were significantly different in concentration groups (*p* < 0.05, n = 9). Data are presented as means ± SD.

## Data Availability

Data are available from the authors.

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
