# Peer review of "Integration of Multi-Omics, Histological, and Biochemical Analysis Reveals the Toxic Responses of Nile Tilapia Liver to Chronic Microcystin-LR Exposure"

_toxins, 2024, doi:10.3390/toxins16030149_

Round 1

Reviewer 1 Report

Comments and Suggestions for Authors

The article titled "The toxic effects and potential mechanisms in livers of Nile tilapia exposed to environmental concentrations of microcystin-LR based on transcriptomics and metabolomics” is about the toxic effect that chronic exposure to MC-LR has on Nile tilapia liver using histological and biochemical determinations and investigates the mechanisms responsible for its toxicity using multi-omics techniques and gene expression. This is a complete and innovative article, however, some modifications are necessary.

- In the "Introduction", it would be appropriate that authors provide more information about the mechanisms responsible for MC-LR toxicity. As well as providing data on permitted concentrations of MC-LR indicated in the official guidelines. 

- What would be the human impact of consuming tilapia contaminated with MC-LR at the concentrations used in the study?. 

- Why have you decided to evaluate concentrations higher than those that can be found in nature?. It is therefore normal to observe alterations in the liver, not?. 

- In some figures legends are not read well, increase quality of images. As well as the words in any of the images by the background colors.

- A deeper discussion of the results obtained and their comparison with others already available in the scientific literature would be necessary.

Author Response

Comments 1: 

The article titled "The toxic effects and potential mechanisms in livers of Nile tilapia exposed to environmental concentrations of microcystin-LR based on transcriptomics and metabolomics” is about the toxic effect that chronic exposure to MC-LR has on Nile tilapia liver using histological and biochemical determinations and investigates the mechanisms responsible for its toxicity using multi-omics techniques and gene expression. This is a complete and innovative article, however, some modifications are necessary.

Response 1:

Thank you very much for your affirmation and comments, and offering us an opportunity to improve our work. We have tried our best to make sure all mistakes are corrected adequately according to your precious comments and suggestions.

Comments 2:  

In the "Introduction", it would be appropriate that authors provide more information about the mechanisms responsible for MC-LR toxicity. As well as providing data on permitted concentrations of MC-LR indicated in the official guidelines.

Response 2

Thank you very much for your precious comments and suggestions.

We have reformulated the "Introduction" according to your suggestions by providing more information about the mechanisms responsible for MC-LR toxicity. Nevertheless, there were no data on permitted concentrations of MC-LR about aquatic animals. Although the concentration of microcystin in drinking water and recreational water for human should not exceed 1 μg/L according to the World Health Organization. And these data seem not appropriate for the manuscript, as it is mainly about the toxicity of MC-LR on fish.

Comments 3:  

What would be the human impact of consuming tilapia contaminated with MC-LR at the concentrations used in the study?.

Response 3:

Thank you very much for your comments.

We agree with this comment that it is very important to explored the impact of consuming tilapia contaminated with MC-LR. Therefore, in fact, we comprehensively investigate the effects of chronic exposure to environmental concentrations of MC-LR on muscle quality and residual risk of Nile tilapia. The manuscript is submitted to another food-related journal (Food chemistry). Based on the principle that the same data cannot be reused in different articles. Thus, this study did not include data about the food safety risks from tilapia contaminated with MC-LR. Nevertheless, it should be feasible to mainly investigate effects of MC-LR exposure on the health of a specific organ of fish according to previous studies, such as “Parental Transfer of Microcystin-LR-Induced Innate Immune Dysfunction of Zebrafish: A Cross-Generational Study” , “Microcystin-RR exposure results in growth impairment by disrupting thyroid endocrine in zebrafish larvae” and “Chronic Microcystin-LR Exposure Induces Abnormal Lipid Metabolism via Endoplasmic Reticulum Stress in Male Zebrafish”.

Comments 4:  

Why have you decided to evaluate concentrations higher than those that can be found in nature?. It is therefore normal to observe alterations in the liver, not?

Response 4: Thank you very much for your precious comments. The ambient levels of MC-LR was very low in nature. Only during the outbreaks of cyanobacterial blooms, the concentrations of MC-LR could researched 10 ug/L, even 30 ug/L in some occasions. Thus, the alterations in the liver of exposed to environmental concentrations of MC-LR should be induced by MC-LR exposure.

Comments 5:  

In some figures legends are not read well, increase quality of images. As well as the words in any of the images by the background colors.

Response 5:

Thank you very much for your precious comments and suggestions.

We have corrected the figures legends and words in the images according to your suggestions.

Comments 6:  

A deeper discussion of the results obtained and their comparison with others already available in the scientific literature would be necessary.

Response 6:

Thank you very much for your precious comments and suggestions.

We have discussed the results deeper by comparing with more available literature according to your suggestions.

Reviewer 2 Report

Comments and Suggestions for Authors

The title should be abbreviated and, why did authors choose to add omics in the title but did not add the other used approaches (histology and biomarkers)?...

In table 1 , and the rest of tables and graphs: I would suggest to start upper-script letters in Control group as "a", and follow up to b/c/d... in the M30 group.

Line 186- remove "etc..."

The full list of liver metabolites per treatment should be available (eventually in repository). The list of differential metabolites in liver should be presented as well. Where did the "muscle" data come from?! There is a list of differential metabolites from muscle in Supplementary material....

Comments on the Quality of English Language

Line 6 "pousinous" to whom? "... "produced in potentially harmful concentrations" would be more adequate

line 10 - remove "chronic", not needed here

line 11 - "histological and biochemical" do not match title...

line 12 - biochemical again... 

line 26  "bail acids" instead of bile acids?

line 36 - "are easier exposed", correct English

line 75: in "different letters were significantly different", add "between concentration groups".

- Remove double spacing along document (e.g.: line 67, 74, ...)

Author Response

Comments 1: 

The title should be abbreviated and, why did authors choose to add omics in the title but did not add the other used approaches (histology and biomarkers)?...

Response 1:

Thank you very much for your precious comments and suggestions and offering us an opportunity to improve our work.

We have corrected the title in revised manuscript according to your suggestions. The title is more abbreviated and comprehensive compared with the previous one.

Comments 2: 

In table 1 , and the rest of tables and graphs: I would suggest to start upper-script letters in Control group as "a", and follow up to b/c/d... in the M30 group.

Response 2:

Thank you very much for your precious comments and suggestions.

We understood that your precious suggestions (start upper-script letters in Control group as "a", and follow up to b/c/d... in the M30 group) may be more conductive to demonstrating the difference between the control and other groups. Nevertheless, after reviewing lots of relevant literature, we found that most studies tended to use traditional methods of marking letters to show the difference among different groups, namely “First, arrange all the averages from largest to smallest, and then mark the largest average with the letter a; And compare this average with the following averages. If the difference is not significant, the letter a will be marked. Until an average that is significantly different from it is marked with the letter b, c...”. We don't mean to offend. We are just worried about confusing readers if a different marking method was used in this manuscript. If it is still necessary, there is no doubt that we will corrected them.  

Comments 3: 

Line 186- remove "etc..."

Response 3:

Thank you very much for your comment.

We have deleted it in revised manuscript.

Comments 4: 

The full list of liver metabolites per treatment should be available (eventually in repository). The list of differential metabolites in liver should be presented as well. Where did the "muscle" data come from?! There is a list of differential metabolites from muscle in Supplementary material....

Response 4:

Thank you very much for your precious comments and suggestions.

We must apologize for our carelessness. A list of differential metabolites in supplementary material were from liver rather than muscle. We have corrected it in supplementary material. Besides, because the full list of metabolites were not requested to be deposited in repository in previous studies associated with metabolomics published in Toxins, such as “High-Resolution Magic Angle Spinning (HRMAS) NMR Identifies Oxidative Stress and Impairment of Energy Metabolism by Zearalenone in Embryonic Stages of Zebrafish (Danio rerio), Olive Flounder (Paralichthys olivaceus) and Yellowtail Snapper (Ocyurus chrysurus)”, “Integrated Metabolomics and Lipidomics Analysis Reveals Lipid Metabolic Disorder in NCM460 Cells Caused by Aflatoxin B1 and Aflatoxin M1 Alone and in Combination”, and “Transcriptomic and Metabolomic Analyses of the Response of Resistant Peanut Seeds to Aspergillus flavus Infection”. Meanwhile, the full list of metabolites were also not requested to be deposited in repository in our previous studies associated with metabolomics, such as “Effects of heat stress on the chemical composition, oxidative stability, muscle metabolism, and meat quality of Nile tilapia (Oreochromis niloticus)” and “The effect of nitrite and nitrate treatment on growth performance, nutritional composition and flavor-associated metabolites of grass carp (Ctenopharyngodon idella)” Thus, unlike the transcriptome, we didn’t realize that we need to save the raw data of metabolomics. Nevertheless, we submitted the table of differential metabolites in the Nile tilapia liver in supplementary file, which displayed the most important information of metabolomics.

Comments 5: 

Comments on the Quality of English Language

Line 6 "pousinous" to whom? "... "produced in potentially harmful concentrations" would be more adequate

line 10 - remove "chronic", not needed here

line 11 - "histological and biochemical" do not match title...

line 12 - biochemical again...

line 26  "bail acids" instead of bile acids?

line 36 - "are easier exposed", correct English

line 75: in "different letters were significantly different", add "between concentration groups".

- Remove double spacing along document (e.g.: line 67, 74, ...)

Response 5:

Thank you very much for your precious comments and suggestions. We have corrected the mistakes about English language as described below.

Line 6: We have reformulated the sentence according to your previous suggestions.

line 10: We have removed "chronic" in line 10.

line 11: We have corrected the title in revised manuscript according to your suggestions.

line 12: After correcting the title in revised manuscript, “Biochemical” should be a appropriate word .

line 26: We have corrected the word in revised manuscript.

line 36 : We have reformulated the phrase in revised manuscript.

line 75: We have corrected the sentences in revised manuscript.

- Remove double spacing along document (e.g.: line 67, 74, ...)

We have rechecked the manuscript and remove double spacing along document.

Round 2

Reviewer 2 Report

Comments and Suggestions for Authors

I consider that the paper in the present form should be accepted for publication. Congratulations for the work.
[ Additional comment: the transparency and availability of data is one of the most important factors to take into account in Science, given its role to all the society. The authors should have this in mind and act accordingly. Keeping raw for a considerable amount of time and making it available to the research community when needed are highly recommended.]